# Comparison of Efficacy and Safety of Cryotherapy in Chemotherapy-Induced Peripheral Neuropathy Between Upper and Lower Extremities: Results from the Randomized CROPSI Study

**DOI:** 10.3390/cancers17111748

**Published:** 2025-05-23

**Authors:** Miriam Emmelheinz, Christian Marth, Daniel Egle, Katharina Leitner, Carmen Albertini, Samira Abdel Azim, Barin Feroz, Laura Strobel, Christine Brunner

**Affiliations:** Department of Obstetrics and Gynecology, Medical University of Innsbruck, 6020 Innsbruck, Austria; miriam.emmelheinz@i-med.ac.at (M.E.);

**Keywords:** breast cancer, taxane, chemotherapy, chemotherapy-induced peripheral neuropathy, cryotherapy

## Abstract

This study investigated the efficacy of cryotherapy between the upper and lower extremities. Temperature measurements as well as neurological tests were conducted during chemotherapy and follow-up. The results suggest that currently available devices for continuous cooling for cryotherapy are significantly more effective on the upper extremity. They also prove that cryotherapy is a safe and effective method to cool extremities to prevent chemotherapy-induced polyneuropathy. Currently, there is still research needed to improve cooling of patients’ lower extremity. Also, larger multicenter studies with standardized tests and questionnaires are needed to assess all possible impacts.

## 1. Introduction

The incidence of gynecological cancer is reported to be 30.3 per 100,000 in the year 2022 [1]. Simultaneously, over recent years, the overall survival of gynecological cancer has drastically improved. As a result, the prevalence and the impact of long-term (side) effects on quality of life (QoL) is gaining significance in treatment planning [2]. Despite constant new developments in cancer treatment, taxane-based chemotherapy (CT) is still often part of therapy for gynecological cancers [3,4]. The clinically most relevant side effect of taxane-based CT is chemotherapy-induced peripheral neuropathy (CIPN) [5,6,7]. This leads to CIPN affecting millions of individuals globally each year [8]. CIPN can heavily impact patients’ long-term QoL. CIPN is typically dose-dependent and distributed symmetrically in a “glove and stocking” manner [9,10,11]. The symptoms include sensory and motor deficits, which may necessitate early discontinuation or dose reduction of CT [5,6,12]. Predictive factors for CIPN are age over 65, diabetes, obesity, prior neuropathy and lifestyle factors like smoking, alcohol consumption and inadequate physical activity [13].

Small studies reported successful prevention of CIPN or reduction in its symptoms by cryotherapy [14,15,16,17,18,19]. Of these studies, two reported on applied cryotherapy only to the upper extremity (UEX) [15,19]. Two metanalyses acknowledged only whether cryotherapy was used solely on upper or on both (upper and lower) extremities [16,18]. Another study by Shigematsu et al. did not separately evaluate the efficacy of upper versus lower extremity (LEX) [14]. Only Hanai et al. distinguished between the extremities regarding the performance in neurological testing [17].

Many studies linked CIPN to depression, anxiety and psychological stress [20]. Furthermore, a more severe CIPN in the lower extremity is associated with a higher risk of falls [21,22]. In a study published by Winters-Stone et al., 47% of women presented with CIPN symptoms six years after treatment, which led to worse function, greater disability and more falls [23]. These results show why prevention of CIPN is important for women’s long-term QoL. They also highlight the importance of preventing CIPN not only in the upper extremity, but also the lower extremity.

The aim of our study was to evaluate whether current continuous cooling devices for cryotherapy are more effective in cooling the upper compared to the lower extremity. We hypothesized that cryotherapy would be more effective in the upper extremity due to various reasons, like blood flow and available cooling procedures. We therefore assessed the temperature changes in each extremity. In addition, we performed neurological evaluations to examine potential sensory deficits. Also, we documented patients’ nail changes, as these often impact patients’ self-esteem.

## 2. Materials and Methods

### 2.1. Study Design

Between May 2020 and January 2023, the CROPSI study was conducted at the Department of Obstetrics and Gynecology at the Medical University of Innsbruck. The primary endpoint was to evaluate if combining cryotherapy and compression could have a mutually reinforcing effect and lead to higher efficacy in the prevention of CIPN. On the upper extremity, patients received cryotherapy or cryocompression depending on their randomization. Due to lack of compression techniques for the LEX during study planning, cryotherapy was applied in both groups. At the commencement of CT (T0), after CT (T1) and twice during follow-up, which lasted up to 9 months (T2 and T3), a wide range of tests and assessments was performed. These tests included temperature measurements and neurological tests. Before as well as after all CT sessions, temperature measurements were conducted.

The study was approved by the local ethics committee and registered as a clinical trial (NCT04632797). Prior to enrollment, written informed consent was obtained from all patients.

### 2.2. Patients

All participants of the CROPSI study who were randomized to receive cryotherapy (C) on their UEX and LEX were included. Randomization was carried out by staff in our clinical study center electronically. Therefore, a total of 97 individuals were enrolled in this subgroup analysis. The results of the CROPSI study regarding the comparison of cryotherapy alone to a combination of compression and cryotherapy (cryocompression, CC) on the upper extremities during chemotherapy treatment have already been published [24].

Inclusion criteria required participants to be at least 18 years old, have a diagnosis of breast or another type of gynecological cancer, and have completed at least three cycles of taxane-based chemotherapy (administered as neoadjuvant, adjuvant, or palliative treatment). Exclusion criteria included existing polyneuropathy of grade 2 or higher, neuralgia, metastases in the bones or soft tissues of the hands or feet, Raynaud’s syndrome, intolerance to cold, peripheral arterial ischemia, or hand-foot syndrome.

During study planning, we assumed that cryotherapy might be more efficient on the upper compared to the lower extremity. Hence, we designed each individual to be their own control. Therefore, only patients with complete assessments at all four points in time were included.

### 2.3. Cooling Procedures

Cryotherapy began 30 min before the administration of taxane-based chemotherapy. It was continued throughout the entire chemotherapy session and for an additional 30 min afterward. The cooling process was carried out using Hilotherm^®^ (Argenbühl-Eisenharz, Germany) cooling devices (as illustrated in Figure 1) (following a recently described method [25]. The devices were set to maintain a temperature between 10 and 12 °C.

### 2.4. Study Endpoints

The primary outcome was the efficacy of temperature reduction of the cryotherapy compared between UEX und LEX. The temperature of the fingertip and big toe was measured before and after every cryotherapy session with the Exergen TemporalScannerTM Thermometer TAT 500^®^.

The secondary outcome of this study was the neurological outcome compared between the upper and lower extremities as well as the safety of cryotherapy. To examine potential sensory deficits, two neurological tests were conducted by blinded assessors. Staff that randomized or administered the cryotherapy devices and gloves did not carry out assessments.

The Rydell–Seifer test and Semmes–Weinstein monofilament test were used in this study [14,18]. The Rydell–Seifer test provides a score evaluating sensory deficits that ranges from 0 to 8. A score of 8 is interpreted as no deficits in recognizing vibration. The Semmes–Weinstein monofilament test evaluates light touch sensation. The score ranges from 3 to 19, 3 representing the smallest prick. Therefore, a high score in the Semmes–Weinstein test correlates with increased sensory loss. The right extremity was used for all assessments and tests except for those with a comorbidity, such as a prior injury, that could impact the test results.

In addition, photographs of the patients’ fingernails were taken by a professional photographer at each checkup. Then, three specialists evaluated nail changes independently using the CTCAE scale (grade 1 = discoloration, ridging or pitting; grade 2 = partial or complete loss of nail(s)) [26].

### 2.5. Statistical Analysis

Analyses were performed using SPSS version 29 (IBM Corp., Armonk, NY, USA). In our analysis, we employed non-parametric tests to evaluate our data. Whenever appropriate and feasible, we utilized paired tests to account for the correlation between matched pairs of observations. The specific non-parametric test selected for our analysis was the Mann–Whitney U test for independent samples.

## 3. Results

### 3.1. Study Population

A total of 97 patients were enrolled in this study (Figure 2). This study evaluated patients who completed their CT with cryotherapy as well as all assessments. Patient characteristics are shown in Table 1. Patients’ median age was 49.5 years and the majority of our cohort were breast cancer patients.

### 3.2. Comparison of Efficacy of the Two Locations

Efficacy was measured by comparing the changes in temperature by extremity; the results are shown in Table 2. On the upper extremity, the median cooling effect was

12.5 °C when measured at the index finger, compared to 9.6 °C on the big toe (*p* < 0.001). Therefore, cooling was significantly more effective on the upper extremity.

### 3.3. Sensory Tests

Regarding the sensory test, all participants scored better on their upper compared to their lower extremity. For the results, see Table 3.

### 3.4. Nail Changes

The nail changes are shown in Table 4. Complete photo documentation was obtained from 21 individuals. Patients in general never developed significant nail changes on their hands. On their feet, patients’ nails changes declined after completion of chemotherapy.

### 3.5. Adverse Events

Regarding side effects, there was no significant difference between upper and lower extremities. No severe side effects were reported for either location. We distinguished between redness and blistering; the results are shown in Table 5.

## 4. Discussion

In this study, temperature measurements showed a higher efficacy of cryotherapy in cooling the upper compared to the lower extremity, respectively. On the UEX, the cryotherapy achieved a cooling effect of 12.5 °C compared to 9.6 °C on the LEX (*p* < 0.001). A study by Ran et al. showed that the efficacy of cryotherapy in preventing CIPN correlates with the achieved cooling temperature, which emphasizes the importance of aiming at a low temperature [27]. In our study, the more efficient temperature reduction in the UEX may have been caused by the fitting of the cooling devices, as the gloves covered the hands up to the forearm, compared to the socks, which did not cover the ankles. In our clinical observation and according to feedback from patients, the lower coverage may have contributed to the lower temperature reduction. Improvements of continuous cooling devices, like more coverage up to the ankles, might be able to improve the efficacy.

Another possible reason is the significantly lower starting temperature for the LEX. The LEX might have profited from a lower set temperature from the cooling devices from the start. The anatomical and physiological differences between the extremities might have also affected cryotherapy efficacy. A study by Stoner et al. showed that blood flow to the feet is more variable and more affected by ambient temperature [28]. Gatt et al. found that under resting conditions, toes were up to 5 °C cooler than fingers [28]. The authors discussed poorer distal perfusion and greater heat loss as the main reasons for this phenomenon. In addition, gravity might lead to lower perfusion pressure in LEX. These factors could to some extent explain the lower starting temperature for the LEX and the less efficient cooling temperature in the LEX. The causes of the significantly lower starting temperature and difference in effectiveness could not be clearly determined in our cohort. This remains an important clinical question for studies with larger patient populations in order to minimize the risk of bias.

Depending on the source, the incidence of CIPN ranges between 70 and 97% after taxane-based CT [11,16,18,25,29]. A study by Visovsky et al. showed that the potential for injury and disability from falls is a considerable concern for women with breast cancer and persistent CIPN [22]. In a study conducted by Winters-Stone et al. women with CIPN reported significantly more disability and 1.8 times higher risk of falls compared to women without CIPN [23]. Similarly, Bao et al. published results where the severity of CIPN was associated with a higher risk of falls [21]. Additionally, CIPN has been linked to insomnia, anxiety and depression by various studies [21]. These studies demonstrate that the prevention of CIPN on all extremities should be taken very seriously when treating patients with taxane-based CT. CIPN has a profound impact on patients’ lives and can affect outcomes if dose reductions in CT become necessary because of CIPN, and may thereby lead to inferior survival [30].

In addition, the symptoms often persist after the completion of CT. This was observed in studies by Winters-Stone et al. and Flatters et al. [23,31]. Flatters et al. observed symptoms of CIPN in 30% of patients six months after the completion of chemotherapy. Winters-Stone et al. did so in 47% of women six years after the completion of CT [23]. The longevity of the CIPN symptoms impacts rehabilitation and the return to productivity [32].

As of now, there is no approved drug for the prevention of CIPN [33]. In the ASCO update of the guidelines for the prevention of CIPN, there is currently no recommendation for agents for the prevention of CIPN. The only agent with appropriate evidence with limited benefit for the treatment of CIPN is duloxetine [34]. Commonly used treatment methods consist of a combination of analgesics, physical therapy and psychosocial interventions [11]. The ESMO guidelines similarly state that currently, no positive recommendation can be given for any agents for the prevention of CIPN [35].

Regarding non-pharmacological interventions for the prevention of CIPN, there have been studies using cryotherapy, compression and cryocompression. The latest NCCN guidelines for invasive breast cancer added a recommendation to consider cryotherapy for hands and feet when treating patients with taxane-containing CT [36].

Cryotherapy can be applied using frozen gloves/socks or a continuous cooling device, like in our study. Out of four studies using frozen gloves and socks, only one study monitored temperature. Chitkumarn et al. changed the gloves and socks every ten to fifteen minutes, whereas the other studies changed them every 45 min or did not provide context [14,15,17,19]. Considering that, in the study with temperature monitoring, frozen medical equipment was changed at least three times more often, it seems likely that in other studies, more fluctuation of temperature took place. In addition, in the study by Beijers et al., 34% of patients discontinued the use of frozen gloves, mainly due to discomfort [19]. In our study, using continuous cooling devices, only three discontinuations (3%) in the cryotherapy group were due to the patient’s decision. Continuous cooling devices provide constant temperature and monitoring, whereas frozen gloves and socks have fluctuating temperatures.

A similar conclusion was drawn in a systematic review by Tai et al. [8]. The authors state that patient discomfort often stems from inappropriate regional cooling with excessively low temperature. They also mention that as a reason for continuous flow cooling methods being better tolerated by patients [8].

The limiting factors for continuous cooling devices include acquisition cost, the need for trained professionals and the need for space in clinics. In addition, the cooling before and after chemotherapy may lead to patients spending more time in the outpatient clinic, which may place additional strain on the already tight schedule. All these practical challenges make wide implementation in patient care difficult. More simple and affordable alternatives should be investigated to find the best method for patients from a medical point of view as well as feasibility in real word scenarios.

Regarding compression alone, there are currently mixed results. A study published by Accordino et al. compared three study arms: compression therapy with garments, cryotherapy with frozen gloves/socks and placebo. In their study, compression therapy was the most effective intervention to prevent CIPN [37], whereas the study by Kontani et al. could not show a significant difference between their intervention group receiving compression with one-size-smaller surgical gloves and their blinded control with normal-sized surgical gloves [38]. A study by Michel et al. also used surgical gloves and compared them to frozen gloves. In this study, there was no significant difference between the cryotherapy and the compression group, and both methods significantly reduced the risk of high-grade CIPN [39]. Like cryotherapy, compression can be applied using different methods, either extra small surgical gloves or continuous compression. Since compression therapy with gloves is cost-effective, simple, and space-saving, it offers many advantages that facilitate its implementation in everyday clinical practice.

A study by Bandla et al. suggested that a combination of cryotherapy and compression might lead to an even more successful prevention of CIPN [40]. The data published by our study group could not confirm these results, as there was no significant difference between the cryotherapy and cryocompression group regarding incidence of CIPN [24].

ESMO guidelines emphasize the importance of early detection of CIPN, and baseline and ongoing clinical evaluation [35]. This highlights the main limitation of all these studies, because the lack of standardized testing, scores and questionnaires makes the comparison of results difficult. A study analyzing assessment methodologies for CIPN concluded that CIPN incidence and prevalence could be confounded by disagreement between assessment modalities [41]. In the future, a standardized testing procedure for CIPN regarding tests, scoring systems and questionnaires is necessary to make research comparable.

In our study, all patients scored significantly better on their neurological test on the UEX compared to their LEX. This was to be expected, but it was interesting that the ratio of the vibration test of upper to lower extremity developed in favor of the upper and then recovered. This result may support the hypothesis that the prevention of symptoms of CIPN was more effective in the extremity with more temperature difference.

Interestingly, the nail changes on the LEX showed a phenomenon which has been described in literature as coasting, where symptoms first worsen after the completion of CT before they improve again [6,35].

The limitations of this study are the lack of a questionnaire which distinguished between the extremities. This study would have highly benefited from extremity-specific questionnaires, as these could determine if the statistically significant difference in temperature reduction leads to less effective prevention of CIPN. During study planning, the decision was made to prioritize QoL with the questionnaires. We did not want to discourage or demotivate patients with overly extensive questionnaires, as this could have led to a lower response rate. Future studies should include extremity-specific questionnaires. Currently, the literature regarding the correlation between the temperature reduction and reduction in CIPN is very scare. Therefore, it is unclear whether a temperature difference of 2.9 °C, as found in our study, is not only statistically but also clinically significant. This is an interesting topic for further research. Regarding nail changes, complete photo documentation was only obtained from less than half of our patients for various reasons. Some patients declined photos during the study, whereas others wore nail polish. It is not possible to rule out whether this caused a bias. It is possible that patients with worse nail changes might have been more likely to decline photos. Even though patients were asked to not wear nail polish and participate in the photo documentation, it is evident that better patient education is needed to achieve more complete results. Patient education should emphasize the importance of photo documentation. And the process of participating in photo documentation should be simple and not time-consuming.

The biggest strength of this study is being one of the first to conduct all neurological exams as well as temperature measurement individually for UEX and LEX and the relatively large patient cohort. Since factors like the method of cryotherapy, patients’ characteristics, combination and scheduling of agents can impact incidence of CIPN, larger multicenter studies with standardized tests and questionnaires are needed to further investigate the most successful methods and possible influences on the prevention of CIPN. Furthermore, our study made evident that there is a need to improve lower extremity continuous cooling devices.

## 5. Conclusions

Our study suggests that currently available devices for continuous cooling for cryotherapy are significantly more effective in temperature reduction in the upper extremity. In this study, cryotherapy was safe and effective to cool patients’ extremities.

## Figures and Tables

**Figure 1 cancers-17-01748-f001:**
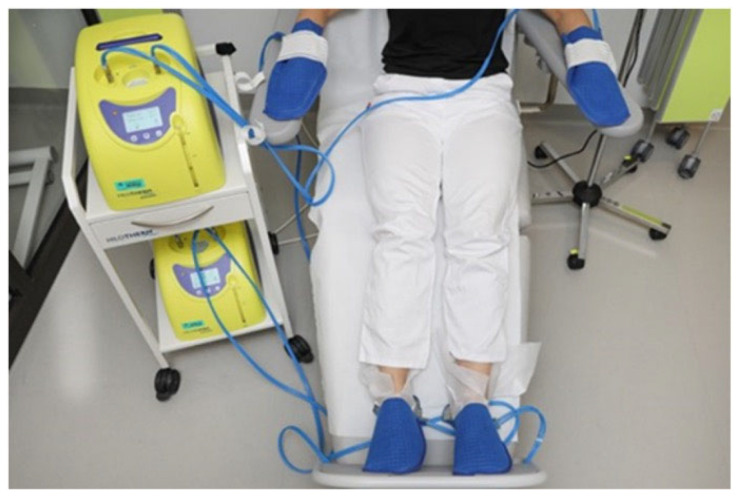
The cooling process.

**Figure 2 cancers-17-01748-f002:**
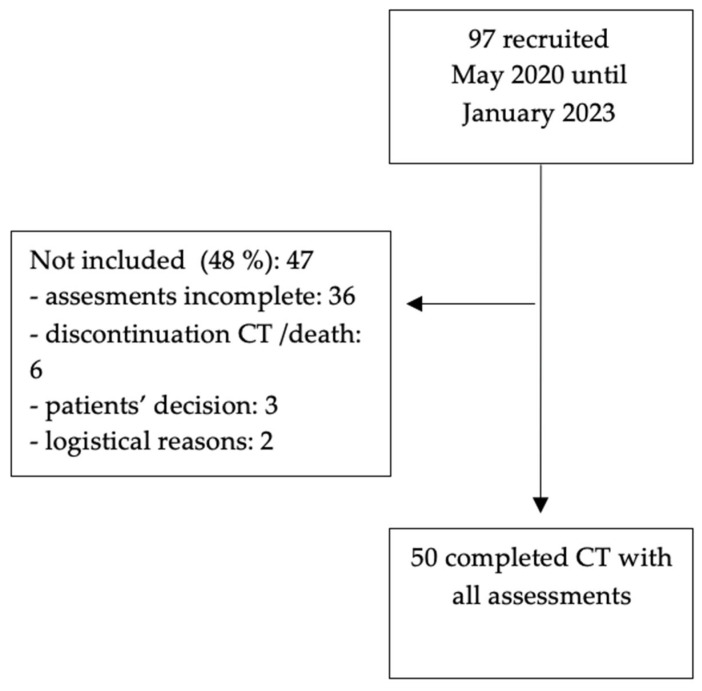
Consort diagram. CT = chemotherapy.

**Table 1 cancers-17-01748-t001:** Patient characteristics.

	n = 50
Age (years): median (range)	49.5 (26–78)
Menopausal status n (%)	
premenopausal	27 (54)
postmenopausal	22 (44)
unknown	1 (2)
BMI: median (range)	23.7 (18.4–38.4)
Tumor features n (%)	
breast cancer	40 (80)
ovarian cancer	8 (16)
other	2 (4)
Chemotherapy regimen n (%)	
Taxane mono	6 (12)
Taxane + anthracycline	32 (64)
Taxane + platinum	12 (24)

**Table 2 cancers-17-01748-t002:** Cooling time and temperature.

	UEX (n = 50)	LEX (n = 50)	*p*-Value
Median Cooling time in minutes (range)	136 (90–285)	136 (90–285)	
Median temperature index finger compared to big toe (°C)			
Median before cooling (range)	36 (27.9–37.3)	29 (24–36.3)	
Median after cooling (range)	23 (16.6–35.9)	21 (15.9–35)	
Median differences (range)	−12.5 (−18.7–(−2.7))	−9.6 (−20.3–(−2))	<0.001

UEX = upper extremity; LEX = lower extremity.

**Table 3 cancers-17-01748-t003:** Neurological tests.

	RS UEX	RS LEX	Ratio UEX/LEX	SW UEX	SW LEX	Ratio UEX/LEX
T0 (range)	8 (5–8)	7 (2–8)	1.1	3 (3–8)	8 (3–15)	0.4
T1 (range)	7 (3–8)	5 (0–8)	1.4	3 (3–13)	9 (3–19)	0.3
T2 (range)	8 (2–8)	6 (0–8)	1.3	4 (3–10)	8 (3–19)	0.5
T3 (range)	7.75 (0–8)	6.5 (0–8)	1.2	4 (3–11)	8 (3–17)	0.5

RS = Rydell–Seifer vibration test; SW = Semmes–Weinstein monofilament test; T0 = start of CT; T1 = end of CT; T2 = 3 months after CT; T3 = 6–9 months after CT; UEX = upper extremity; LEX = lower extremity.

**Table 4 cancers-17-01748-t004:** Nail changes.

Grade of Nail Change n = 21	UEX	LEX
T0 (range)	0 (0–1)	0 (0–2)
T1 (range)	0 (0–1)	0 (0–2)
T2 (range)	0 (0–1)	1 (0–2)
T3 (range)	0 (0–1)	1 (0–2)

T0 = start of CT; T1 = end of CT; T2 = 3 months after CT; T3 = 6–9 months after CT; UEX = upper extremity; LEX = lower extremity (grade 1 = discoloration, ridging or pitting; grade 2 = partial or complete loss of nail(s)) [26].

**Table 5 cancers-17-01748-t005:** Frequency of adverse events.

	UEX n = 50 (%)		LEX n = 50 (%)
Redness hand never	31 (62)	Redness foot never	29 (58)
Redness hand 1–2x	17 (34)	Redness foot 1–2x	20 (40)
Redness hand 3–4x	2 (4)	Redness foot 3x	1 (2)
Blistering hand never	49 (98)	Blistering foot never	49 (98)
Blistering hand once	1 (2)	Blistering foot once	1 (2)

UEX = upper extremity; LEX = lower extremity.

## Data Availability

Data is available from the corresponding author upon reasonable request.

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
