# Peer review of "Comparison of Efficacy and Safety of Cryotherapy in Chemotherapy-Induced Peripheral Neuropathy Between Upper and Lower Extremities: Results from the Randomized CROPSI Study"

_cancers, 2025, doi:10.3390/cancers17111748_

Round 1
Reviewer 1 Report
Comments and Suggestions for Authors
The study addresses an important clinical issue on chemotherapy-induced peripheral neuropathy (CIPN) and evaluates the differential efficacy of cryotherapy between upper and lower extremities.
While the topic is relevant, the manuscript has several weaknesses that need addressing before publication.
I provide detailed comments and constructive feedback below to help the authors improve their manuscript.
- The abstract doesn’t have clarity in stating the primary and secondary outcomes explicitly. The phrase "higher efficacy of cryotherapy in cooling" is vague; specify whether this refers to temperature reduction or CIPN prevention.
- Clearly define primary (temperature reduction) and secondary (neurological outcomes, safety) endpoints.
- The p-value for temperature difference is highlighted, but the clinical significance of a 2.9°C difference is not discussed.
- Briefly state the randomization method and whether assessors were blinded.
- Add a sentence on the clinical implications of the temperature difference observed.
- There’s no mention of the randomization process or blinding, which are critical for interpreting results.
- The logic for comparing upper vs lower extremities is underdeveloped. The introduction focuses broadly on CIPN but does not sufficiently justify why extremity-specific differences matter beyond citing a single study (Hanai et al.).
- The literature review is uneven: it cites small studies and meta-analyses but misses larger trials or recent systematic reviews (e.g., Jordan et al., 2020).
- Strengthen the rationale by discussing anatomical/physiological differences (e.g., blood flow, tissue composition) that may affect cryotherapy efficacy.
- Include recent high-impact studies on CIPN prevention (ASCO guidelines).
- The hypothesis is implied but not explicitly mentioned. Clearly state the hypothesis: "We hypothesized that cryotherapy would be more effective in the upper extremity due to … [specific reasons]."
- The manuscript conflates two studies (CROPSI and the current analysis). It is unclear whether this is a subgroup analysis or a separate cohort.
- With 50/97 patients completing assessments, attrition bias is likely so power analysis is needed
- The temperature setting (10–12 C) is arbitrary; no justification is given for this range.
- The Rydell-Seifer and Semmes-Weinstein tests are described, but their validity/reliability in CIPN are not discussed. Inter-rater reliability for nail changes is mentioned, but no metrics (e.g., Cohen’s kappa) are provided.
- It’s not clear if assessors were blinded to the intervention, which risks bias in neurological evaluations.
- The baseline temperature difference (36C vs 29C) between extremities is extreme and unexplained. This could confound the cooling efficacy comparison. What’s the logic here?
- The ratio of UEX/LEX scores is presented without statistical testing (e.g., paired t-tests). The clinical relevance of these ratios is unclear.
- Data are incomplete (only 21/50 patients), and no statistical analysis is provided (e.g., Fisher’s exact test).
- Redness/blistering rates are descriptive; no statistical comparison between extremities is made.
- The conclusion that cryotherapy is "significantly more effective" on UEX is based solely on temperature, not CIPN incidence (which was not a primary endpoint).
- The discussion doesn’t address why baseline temperatures differed or how this might bias results.
- The comparison to frozen gloves/socks is superficial. The authors should discuss why continuous cooling might differ (e.g., sustained vs. fluctuating temperatures).
- The lack of extremity-specific CIPN questionnaires and high attrition rate are underemphasized.
- In conclusion, the study didn’t measure CIPN prevention directly, however the conclusion implies clinical utility.
- There’s no mention of the need for improved lower-extremity devices, which is a key finding.
- Table 1: include age range too.
Author Response
Reviewer 1:
I provide detailed comments and constructive feedback below to help the authors improve their manuscript.
We sincerely thank the reviewer for the valuable and detailed feedback. We have tried to thoroughly address all comments and incorporate the suggested improvements. All line references refer to the marked version.
- The abstract doesn’t have clarity in stating the primary and secondary outcomes explicitly. The phrase "higher efficacy of cryotherapy in cooling" is vague; specify whether this refers to temperature reduction or CIPN prevention.
The revision has been implemented as suggested by the reviewer (line 21).
- Clearly define primary (temperature reduction) and secondary (neurological outcomes, safety) endpoints.
The improvement was made in accordance with the reviewer’s comment (line 158 - 167).
- The p-value for temperature difference is highlighted, but the clinical significance of a 2.9°C difference is not discussed.
We thank the reviewer for this valuable remark. Currently the literature on the correlation between temperature reduction and reduction of CIPN is extremely scare. Unfortunately, in our study a single CIPN assessment was used for both upper and lower extremity. Therefore, our primary endpoint was the efficacy in temperature reduction. Whether a temperature difference of 2.9°C is not only statistically but clinically significant is an interesting question for further research. We discussed this in our limitations (line 355-358).
- Briefly state the randomization method and whether assessors were blinded.
The missing information was added per the reviewer’s recommendation (line 124 and 165-167) .
- Add a sentence on the clinical implications of the temperature difference observed.
Per the reviewers’ suggestion a sentence was added to the limitation section (line 241-251 and 349-358).
- There’s no mention of the randomization process or blinding, which are critical for interpreting results.
The missing information was added per the reviewers recommendation (line 124 and 165-167) .
- The logic for comparing upper vs lower extremities is underdeveloped. The introduction focuses broadly on CIPN but does not sufficiently justify why extremity-specific differences matter beyond citing a single study (Hanai et al.).
The paragraph lines 76-82 mentions three studies that link CIPN in the lower extremity to higher risk of falls, this underlines why the prevention of CIPN in the lower extremity is important. Furthermore, the paragraph above ( line 68-74 ) mentions that so far not much research has been done regarding the comparison of UEX and LEX, which supports the need for studies like ours comparing both.
- The literature review is uneven: it cites small studies and meta-analyses but misses larger trials or recent systematic reviews (e.g., Jordan et al., 2020).
We thank the reviewer for this remark. If the reviewer is referring to the ESMO guidelines published by Jordan et al. 2020 these are already incorporated in our paper (reference 35 ). Should the reviewer have been referring to a different paper, we would be pleased to incorporate it into our manuscript upon receiving the reference. Furthermore, to strengthen the literature section, we added a
meta-analysis by Hsiu‐Yu Tai (reference 8) .
- Strengthen the rationale by discussing anatomical/physiological differences (e.g., blood flow, tissue composition) that may affect cryotherapy efficacy.
This is a very intriguing question. A thorough discussion of this topic would provide enough material for a dedicated paper. Due to the time constraints of this review and the numerous comments we received, we were only able to address it briefly. However, we have identified a paper that describes blood flow in the feet as being variable and more affected by ambient temperature than in the hands, leading to greater fluctuations in skin temperature of the lower limbs. We also discussed some other possible differences leading to the difference in efficacy. (line 241-251)
- Include recent high-impact studies on CIPN prevention (ASCO guidelines).
Our paper among others discusses the following recent high-impact studies regarding CIPN prevention: Michel et al. (JAMA Oncol. 2025) and Accordino et al. (Breast Cancer Res Treat. 2024) as well as the ASCO and ESMO guidelines. Should an other recent study be missing, we would be pleased to incorporate it into our manuscript.
- The hypothesis is implied but not explicitly mentioned. Clearly state the hypothesis: "We hypothesized that cryotherapy would be more effective in the upper extremity due to … [specific reasons]."
We added the hypothesis to our Introduction (line 86) .
- The manuscript conflates two studies (CROPSI and the current analysis). It is unclear whether this is a subgroup analysis or a separate cohort.
This is a subgroup analysis. All patients from the cryotherapy arm of the CROPSI study were included. We added subgroup analysis to clarify the question (line 125).
- With 50/97 patients completing assessments, attrition bias is likely so power analysis is needed
A fundamental challenge in calculating an appropriate sample size lies in the necessity of specifying the expected effect size. However, estimating the effect size a priori is inherently difficult—particularly in the context of a novel investigation—when no prior empirical data are available, and the study in question is designed as a pilot. Without preliminary findings, any assumptions regarding the magnitude of the effect would be speculative and potentially misleading. At the time of our study’s conception, no comparable data existed in the literature that could have informed a reliable estimation of the effect size for our specific research question. As a result, we deliberately designed this study as a pilot with a limited sample size, with the aim of generating preliminary data that would allow us to estimate the effect size empirically. These initial findings can then serve as a foundation for more accurate sample size calculations in future, larger-scale studies investigating the same or similar phenomena.
- The temperature setting (10–12 C) is arbitrary; no justification is given for this range.
This is an excellent question. The temperature was set to 11°C per the manufacture’s recommendation; however, deviations between 10°C and 12°C were permissible according to our study protocol.
- The Rydell-Seifer and Semmes-Weinstein tests are described, but their validity/reliability in CIPN are not discussed. Inter-rater reliability for nail changes is mentioned, but no metrics (e.g., Cohen’s kappa) are provided.
As we describe in our paper, a major challenge in CIPN research is the lack of a gold standard for tests and questionnaires. At the time the study was designed, the available literature that could have served as guidance was significantly more limited than it is today. The decision to use these two tests was based on a review of the literature and consultations with colleagues in neurology. The idea was that the two tests would measure vibration and sensory deficits and thus capture key symptoms of CIPN.
We thank the reviewer for the insightful question regarding inter-rater reliability. In our opinion, the Cohen’s Kappa test appears to be unsuitable in this context, as it is designed for two raters, whereas in our study the nails were evaluated by three experts. A more appropriate method might be Fleiss’ Kappa. Since this calculation would need to be performed for both the upper and lower extremities at all time points, we would be happy to include these results in the next revision, provided the reviewer agrees that Fleiss’ Kappa is appropriate or suggests a more suitable alternative.
- It’s not clear if assessors were blinded to the intervention, which risks bias in neurological evaluations.
Assessors were blinded to the intervention as staff that randomized or administered the cryotherapy devices and gloves did not carry out assessments. The missing information was added (line 165-167).
- The baseline temperature difference (36C vs 29C) between extremities is extreme and unexplained. This could confound the cooling efficacy comparison. What’s the logic here?
A study by Gatt et al. found that under resting conditions toes were up to 5°C cooler than fingers. They named poorer distal perfusion and greater heat loss as main reasons for this phenomenon. In addition gravity might lead to lower perfusion pressure in the lower extremity. This again provides very interesting topics for further research (line 241-251).
- The ratio of UEX/LEX scores is presented without statistical testing (e.g., paired t-tests). The clinical relevance of these ratios is unclear.
Given the differing baselines, a statistical comparison between the upper and lower extremities did not yield meaningful insights in our opinion. As a result, we opted to illustrate the changes using ratios. Nevertheless, we can provide statistical analyses at each time point if this would improve clarity.
- Data are incomplete (only 21/50 patients), and no statistical analysis is provided (e.g., Fisher’s exact test).
Given the differing baselines, a statistical comparison between the upper and lower extremities did not yield meaningful insights in our opinion. Nevertheless, we can provide statistical analyses at each time point if this would improve clarity.
- Redness/blistering rates are descriptive; no statistical comparison between extremities is made.
Since the percentages are so similar, we initially thought it was unnecessary to report statistical significance, but we are happy to provide it if needed.
- The conclusion that cryotherapy is "significantly more effective" on UEX is based solely on temperature, not CIPN incidence (which was not a primary endpoint).
We clarified the conclusion per the reviewer’s suggestion. (line 380)
- The discussion doesn’t address why baseline temperatures differed or how this might bias results.
We added some possible reasons for the different baseline (line 241-251). But the causes of the significantly lower starting temperature and difference of effectiveness could not be clearly determined in our cohort. This remains an important clinical question for studies with larger patient populations in order to minimize the risk of bias.
- The comparison to frozen gloves/socks is superficial. The authors should discuss why continuous cooling might differ (e.g., sustained vs. fluctuating temperatures).
We incorporated sentences to discuss the differences more thoroughly as per the reviewer’s recommendation (line 397-308, 321-323).
- The lack of extremity-specific CIPN questionnaires and high attrition rate are underemphasized.
We expanded the section discussing the two limitations as per the reviewers recommendation (line 349- 366).
- In conclusion, the study didn’t measure CIPN prevention directly, however the conclusion implies clinical utility.
We clarified the conclusion per the reviewer’s suggestion (line 380).
- There’s no mention of the need for improved lower-extremity devices, which is a key finding.
We included the need for improved lower-extremity devices per the reviewers recommendation ( line 234-237, 374) .
- Table 1: include age range too.
The age range is provided in the brackets behind the median age.
Reviewer 2 Report
Comments and Suggestions for Authors
The manuscript “Comparison of Efficacy and Safety of Cryotherapy in Chemotherapy Induced Peripheral Neuropathy Between Upper and Lower Extremity: Results from the Randomized CROPSI Study”, by Miriam Emmelheinz et al.. The submitted manuscript presents a well-structured, prospective randomized clinical study investigating the comparative effectiveness and safety of cryotherapy applied to the upper versus lower extremities for the prevention of chemotherapy-induced peripheral neuropathy (CIPN) in taxane-treated patients with breast and gynecological cancers. The study stands out for using continuous cooling systems and standardized neurological assessments across multiple time points. The findings are clinically relevant and add to the limited comparative data between upper and lower extremity cryotherapy.
Revisions:
- Explain more clearly why the cooling socks for the lower extremity were less effective, especially because they did not cover the ankles. Suggest how better designs could improve results in future studies.
- Mention that not all patients had complete nail documentation and explain how this might affect the reliability of nail change results. Suggest steps for improving nail photo collection in future trials.
- Point out that the study did not include patient-reported outcomes that focused on each extremity. Recommend using questionnaires that separately measure symptoms in hands and feet in future research.
- Discuss in more detail how the lower extremities had cooler baseline temperatures than the upper ones. Explain how this difference might have influenced the results and suggest ways to adjust for it in future studies.
- Recommend using standardized cryotherapy procedures and scoring systems. Highlight that other studies used different methods, which makes it hard to compare results across studies.
- Discuss the practical challenges of using continuous cooling machines, such as cost, space, and the need for trained staff. In future research, suggest evaluating simpler or more affordable alternatives.
- Add a short explanation about whether the number of patients in the study was enough to detect meaningful differences. This helps readers understand how reliable the findings are.
All the best!
Round 2
Reviewer 1 Report
Comments and Suggestions for Authors
The authors have addressed my comments well.